# Porous borders at the wild-crop interface promote weed adaptation in Southeast Asia

Lin-Feng Li [1,2,3,9], Tonapha Pusadee[4,9], Marshall J. Wedger[3], Ya-Ling Li[1], Ming-Rui Li[1], Yee-Ling Lau[5], Soo-Joo Yap[6], Sansanee Jamjod[4], Benjavan Rerkasem[7], Yan Hao[2], Beng-Kah Song [8] ✉ & Kenneth M. Olsen [3] ✉

High reproductive compatibility between crops and their wild relatives can provide benefits for crop breeding but also poses risks for agricultural weed evolution. Weedy rice is a feral relative of rice that infests paddies and causes severe crop losses worldwide. In regions of tropical Asia where the wild progenitor of rice occurs, weedy rice could be influenced by hybridization with the wild species. Genomic analysis of this phenomenon has been very limited. Here we use whole genome sequence analyses of 217 wild, weedy and cultivated rice samples to show that wild rice hybridization has contributed substantially to the evolution of Southeast Asian weedy rice, with some strains acquiring weed-adaptive traits through introgression from the wild progenitor. Our study highlights how adaptive introgression from wild species can contribute to agricultural weed evolution, and it provides a case study of parallel evolution of weediness in independently-evolved strains of a weedy crop relative.

The emergence of reproductive isolation between domesticated species and their wild relatives is an important but underrecognized component of the crop domestication syndrome[1,2]. Crop species vary widely in their degree of reproductive isolation from wild relatives[3], and this has had major consequences for crop species evolution at both the genomic and phenotypic levels. One important impact is on the extent to which introgression from sympatric wild populations may mitigate the 'cost of domestication'[4,5], i.e., the accumulation of deleterious mutations due to demographic and selective forces imposed during domestication[4,6–9]. Conversely, in crop species where reproductive isolation from wild populations is high, the lack of gene flow can prevent introgression of maladaptive genetic variants and/or undesirable traits that may otherwise enter the domesticated gene pool[10].

Along the spectrum of crop-wild reproductive isolation, Asian rice (*Oryza sativa*) is characterized by an inherently low level of isolation from its wild progenitor, *O. rufipogon*[11]; this porous species boundary is attributable in large part to genetic incompatibility loci that remain polymorphic both within the crop species and in related wild *Oryzas*[11,12]. While the continued reproductive connection with wild rice has utility for germplasm improvement[13], it can also have negative consequences. Here we examine an underexplored negative impact of the low reproductive isolation between domesticated and wild rice: its role in the adaptation of weedy rice (*Oryza* spp.), a feral descendant of Asian rice.

Weedy rice invades rice production areas worldwide, where it aggressively competes with the crop for resources and severely reduces yields[14]. It is among the most problematic agricultural weeds of rice fields in the United States and many other regions[15–17], and its abundance has increased globally with agricultural shifts away from traditional hand-transplanted rice farming to mechanized direct-

[1]State Key Laboratory of Biocontrol, Guangdong Provincial Key Laboratory of Plant Resources, School of Life Sciences, Sun Yat-sen University, Guangzhou 510275, China. [2]Ministry of Education Key Laboratory for Biodiversity Science and Ecological Engineering, School of Life Sciences, Fudan University, Shanghai 200438, China. [3]Department of Biology, Washington University in St. Louis, St. Louis, MO 63105, USA. [4]Department of Plant and Soil Sciences, Faculty of Agriculture, Chiang Mai University, Chiang Mai 50200, Thailand. [5]Department of Parasitology, Faculty of Medicine, University Malaya, Kuala Lumpur, Malaysia. [6]Codon Genomics, Seri Kembangan, Malaysia. [7]Plant Genetic Resources and Nutrition Laboratory, Chiang Mai University, Chiang Mai 50200, Thailand. [8]School of Sciences, Monash University Malaysia, 47500 Bandar Sunway, Selangor, Malaysia. [9]These authors contributed equally: Lin-Feng Li, Tonapha Pusadee. ✉e-mail: song.beng.kah@monash.edu; kolsen@wustl.edu

seeding[18,19]. A growing area of interest has been in understanding the evolutionary mechanisms that underlie the origin and adaptation of weedy rice around the world[20–22]. Most weedy rice strains examined to date appear to have originated directly from local crop varieties through a process of de-domestication (endoferalization). This mechanism is particularly well documented for weedy rice in temperate regions (*e.g.*, United States, Europe and northeastern Asia), where no reproductively compatible wild relatives occur[23–28]. Recent comparative genomic analyses have revealed that these temperate weed strains have evolved through multiple independent de-domestication events in different world regions; these studies have also begun to elucidate the molecular basis of some weediness traits, including seed shattering, seed dormancy and herbicide resistance[28–32]. At the local geographical level, temperate weed populations typically show low genome-wide diversity, consistent with their demographic history of compounded bottlenecks (the rice domestication bottleneck followed by the founding of endoferal weeds)[30].

In contrast to temperate regions, rice production in some areas of tropical Asia occurs in sympatry with the common wild rice progenitor, *Oryza rufipogon*. This region includes portions of southern China, continental and island Southeast Asia, and parts of the Indian subcontinent. *Oryza rufipogon* is interfertile with both domesticated and weedy rice, and while some traits of the wild species would be expected to be maladaptive for weedy rice in the agricultural fields where it specializes (*e.g.*, perenniality and prostrate growth), other wild rice traits are likely advantageous (*e.g.*, seed shattering, seed dispersal structures such as awns, and seed dormancy). In keeping with this expectation, candidate gene and neutral marker studies of weedy rice populations in Southeast and South Asia have indicated that wild rice can contribute to local weed evolution by hybridization and adaptive introgression[33–39]. However, whereas temperate weedy rice has been extensively examined through whole genome sequence analysis[28,30–32], genome resequencing of tropical Asian weedy rice has so far been limited[28]. Consequently, the selective signatures and genome evolutionary dynamics of tropical Asian weedy strains are still unclear. In particular, the evolutionary mechanisms underlying weedy rice adaptation (*e.g.*, the molecular basis of weediness traits) in regions with wild rice remain largely unexplored.

Here we examine the evolution of weedy rice in Southeast Asia, a region of sympatry with wild rice, in comparison to temperate weed stains, to assess the role of wild rice introgression in the evolution and adaptation of these tropical weed strains. Our population genomic inferences reveal that in comparison to weedy rice strains in temperate regions, which appear to be descended almost exclusively from domesticated rice ancestors (either directly via de-domestication or indirectly via crop-weed or weed-weed hybridization)[28,30,40,41], Southeast Asian weedy rice presents a far more dynamic and heterogeneous picture of weedy crop relatives and their mechanisms of adaptation in the presence of reproductively compatible wild species. Our study thereby provides an evolutionary perspective on how adaptive introgression from wild relatives can overcome the 'cost of domestication' in agricultural weeds that are descended from domesticated ancestors.

## Results
### Genome sequencing and SNP identification
We generated 34 new whole genome sequences for Southeast Asian (SEA) rice accessions; these included 31 weed strains (13 Malaysian and 18 Thai) plus two Malaysian *indica* elite cultivars and one Malaysian wild rice sample that were sampled to increase representation of SEA crop and wild rice germplasm. Average coverage for the newly obtained genomes was >20× genome coverage (generated using the Illumina HiSeq 2000 platform). These were analyzed together with 183 previously-published *Oryza* genome sequences comprising the following: 89 cultivated rice accessions representing the five major genetic subgroups within the crop (44 *indica*, 16 *aus*, 10 *tropical*

*japonica*, 14 *temperate japonica* and five *aromatic*); 53 wild rice accessions representing the current geographic distribution of the progenitor species (including 10 accessions of the annual form, *O. nivara*); 38 United States weedy rice strains (18 *indica*-like straw-hull strains, designated US-SH; and 20 *aus*-like black-hull awned strains, designated US-BHA); and three Chinese weedy rice accessions from central China[30,40,42] (Supplementary Data 1). Raw reads were filtered and mapped onto the Nipponbare (*temperate japonica*) reference genome (MSU 6.0 release; http://rice.plantbiology.msu.edu). A total of 33,927,059 raw variants were identified in the 217 accessions, of which 4,718,646 and 8,026,996 were found in Malaysian and Thai weed strains, respectively.

### Phylogenetic and ancestry inferences
To infer evolutionary relationships among the SEA weedy rice strains, we performed phylogenetic analyses based on 1,455,426 homozygous single-nucleotide polymorphisms (SNPs) in MEGA7[43]. Overall clustering patterns among the 183 previously published samples are consistent with known genetic relationships for cultivated and wild rice (Fig. 1 and Supplementary Fig. 1)[30,42]. For SEA weedy rice, despite high morphological variation in the collected samples (which together comprise straw-hull, brown-hull awned, and black-hull awned seed morphotypes; Supplementary Data 1), all newly sequenced strains grouped with SEA *indica* crop varieties and wild rice accessions, although without strong statistical support (BS < 50%). Out of the 31 SEA weedy rice accessions, six Thai and four Malaysian weeds clustered with the Malaysian and Indonesian *indica* varieties (BS < 50%); nine Malaysian weeds grouped most closely with two Malaysian *indica* elite cultivars (BS = 77%) (Supplementary Fig. 1). In contrast, the remaining 12 Thai weeds formed a well-supported group together with SEA wild rice (BS = 73%), indicating a close relationship to the wild species and the possibility of recent hybridization and introgression from wild populations. A similar pattern was also evident in principal component analysis (PCA), where both Malaysian and Thai weeds clustered with their crop ancestor (Supplementary Fig. 2). The

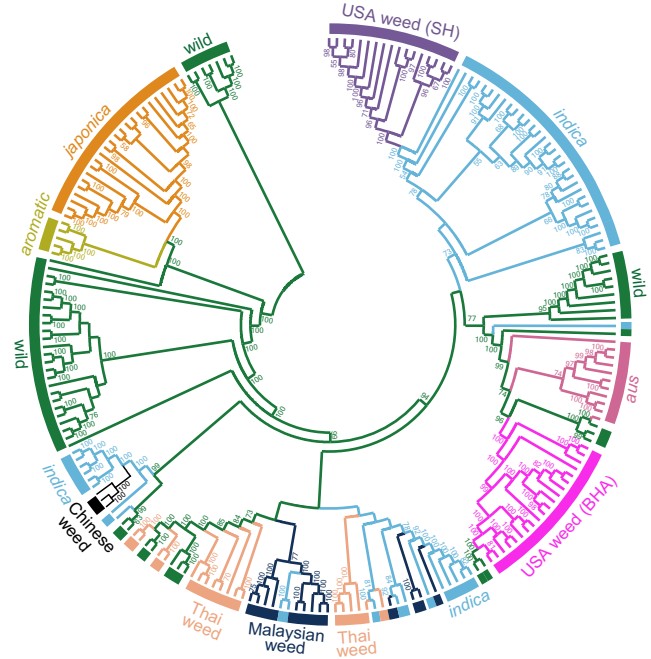

**Fig. 1 | Neighbor-Joining tree of the 217 worldwide rice accessions.** Colors represent different rice groups used in this study. Numbers above each branch indicate bootstrap support values (>50%). Tree topologies were rooted using wild rice Or-IIIa group as the outgroup according to Huang et al. [42]. and Li et al. [30]. Source data are provided as a Source Data file.

population structure of SEA weedy rice is correlated with the geographic distribution of wild rice, in which *indica*-like wild rice is more prevalent in Thailand and less prevalent in the Peninsular Malaysia region[44].

To further examine SEA weedy rice origins, we performed a maximum likelihood estimation of individual ancestries using ADMIXTURE with a reduced SNP dataset consisting of 573,655 loci (see Methods; Fig. 2a, b and Supplementary Data 2). Inferred ancestries for weed strains were broadly congruent with phylogenetic inferences, with the US-BHA weeds grouped with *aus* rice, and all four other weed groups (US-SH, Chinese, Malaysian and Thai weeds) sharing higher similarity with *indica* rice than with any other domesticated group. Consistent with the phylogenetic analysis, more than half of Thai weeds shared some ancestry with SEA wild rice (specifically population group 'Or-I' following the nomenclature of previous study[42]) (11.5-49.1% membership, light blue color at K = 6; Fig. 2a). Three Malaysian weeds also showed some degree of shared ancestry with a genetic population present in *aus* and SEA wild rice (group Or-I) (7.4-13.7% membership, green color at K = 6; Fig. 2a). These genomic features allowed us to divide the SEA weedy rice into two main groups, namely those closely related to *indica* rice ('*indica*-like') and those with evidence of wild rice ancestry ('wild-like').

To examine broad-scale patterns of historic gene flow between wild, cultivated, and weedy rice, we employed the program Treemix[45], which estimates historical relationships among populations that have experienced migration. For all analyzed samples, the single strongest inferred gene flow event was from wild rice into the wild-like subset of Thai weeds (Supplementary Fig. 3). This supports the inference of wild rice ancestry identified for these Thai accessions in the phylogenetic and ADMIXTURE analyses (Fig. 1, Fig. 2a, b). In addition, a weaker gene flow event was inferred from the *aus* cultivated rice lineage to *aromatic* crop

varieties, supporting a previously reported hypothesis that *aromatic* varieties evolved through hybridization of *aus* and *japonica* rice[46,47].

## Genome-wide patterns of nucleotide diversity and heterozygosity

Outside the geographical range of wild rice, weedy rice strains often show a dramatic decrease in nucleotide diversity compared to both wild and cultivated rice, reflecting genetic bottlenecks in the de-domestication process[28,30]. For SEA weedy rice, both the Malaysian and Thai strains exhibited higher nucleotide diversity (π = 0.00405 and 0.00530, respectively) compared to temperate weedy rice strains (π = 0.00221 for US-SH and π = 0.00320 for US-BHA) (Supplementary Data 3). The wild-like Thai weeds (π = 0.00567) showed the highest nucleotide diversity compared to the other four *indica*-like weedy strains (π = 0.00221 to 0.00405), even exceeding the diversity found in *indica* crop varieties (π = 0.00498) (Supplementary Data 3). This elevated genetic diversity is again consistent with introgression from wild rice into the wild-like Thai weed population.

To further assess the relative roles of wild vs. cultivated rice as contributors to the genetic diversity of the different SEA weedy rice strains, we examined the relative numbers of SNPs in weed accessions that were either characteristic of cultivated rice (i.e., detected in cultivated but not wild rice) or characteristic of wild rice (detected in wild but not cultivated rice). Compared to the Chinese weedy rice accessions, which represent the closest relatives of SEA weedy rice occurring outside the range of wild rice (Fig. 1 and Fig. 2), SEA weedy rice showed marked increases in wild-specific SNPs. For example, while the Chinese weeds showed a > 2:1 ratio of crop-specific to wild-specific SNPs, Malaysian weeds and *indica*-like Thai weeds showed approximately equal proportions of crop-specific and wild-specific SNPs, and wild-like Thai weed accessions showed a

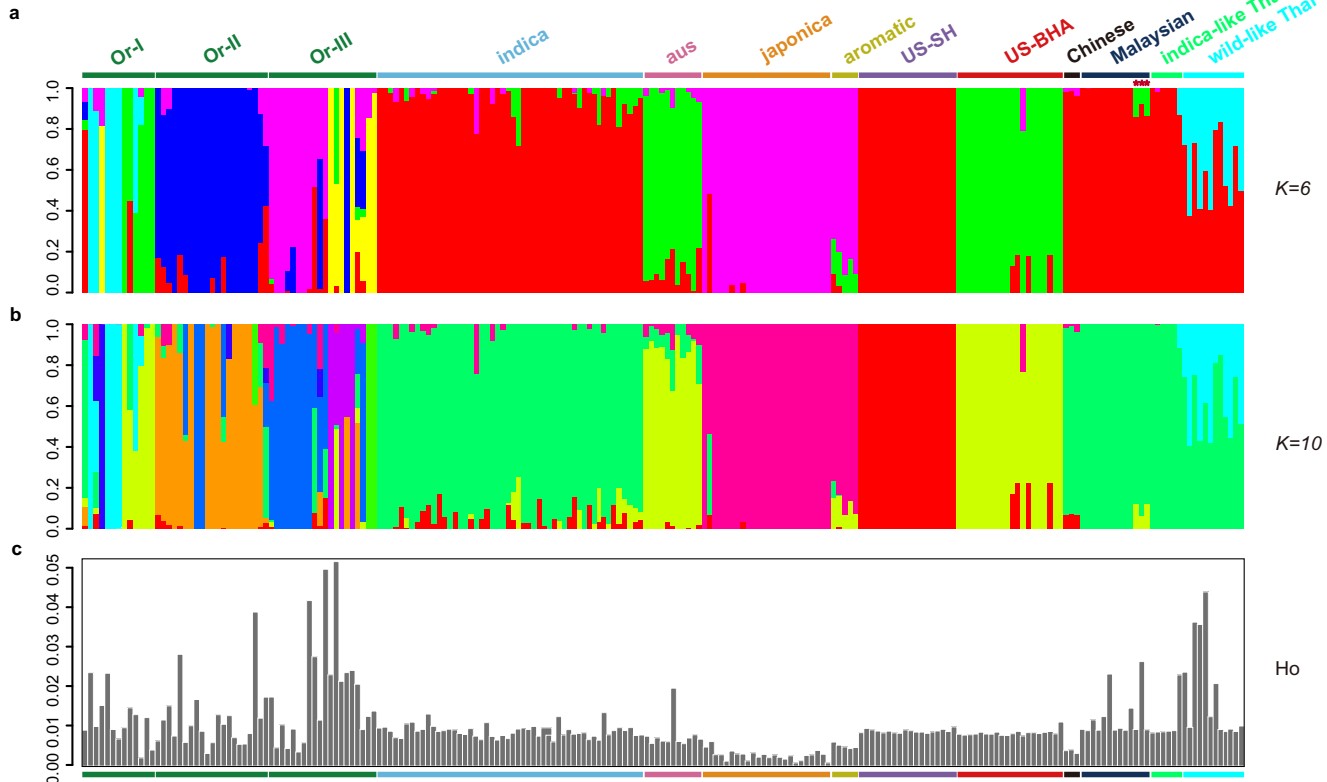

**Fig. 2 | Bayesian genetic assignments and heterozygosity rate for each of the 217 rice accessions.** Each bar represents a single individual. Numbers on the y-axis indicate proportion of ancestry (**a, b**) and heterozygosity rate (**c**) for each rice accession. Characters at the bottom indicate the names of each rice groups. Or-I, Or-II and Or-III indicate *O. rufipogon* (wild rice) subpopulations following the nomenclature of Huang et al.[42]. Source data are provided as a Source Data file.

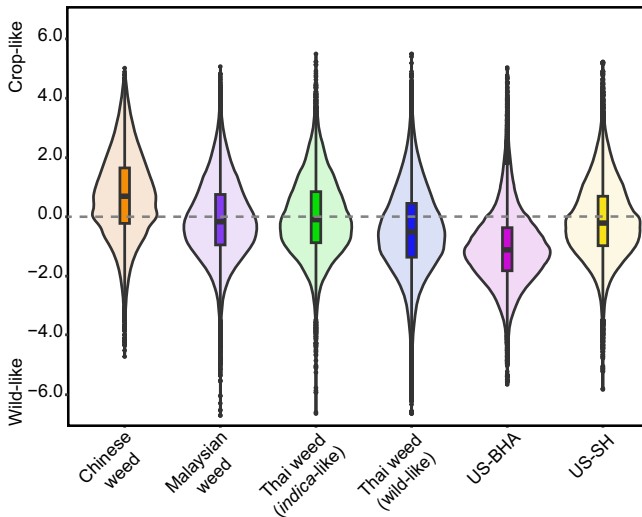

**Fig. 3 | Ratio of crop- and wild-specific private SNPs identified in weedy rice strains.** The gray dashed line indicates equal numbers of crop- and wild-specific private SNPs within the 100-kb sliding window. The box in violin diagrams indicates 95% of the ratio values between crop-specific and wild-specific private SNPs in each of these weedy rice strains. The line within the box indicates the median value for each weedy rice strain. The numbers of private SNPs are included in Supplementary Data 4 and 5. Accession numbers of these weedy rice strains are shown in Supplementary Data 1. Source data are provided as a Source Data file.

2:3 ratio towards an excess of wild-specific SNPs (Fig. 3, Supplementary Fig. 4; Supplementary Data 4 and 5). Malaysian and *indica*-like Thai weeds possess similar ratio of crop- and wild-specific SNPs to US-SH weedy rice (*indica* background), while wild-like Thai weed is more similar to US-BHA weedy rice (*aus* background) (Fig. 3, Supplementary Fig. 4; Supplementary Data 4, and 5). These patterns are consistent with the above inference that wild rice has disproportionately contributed to the genetic composition of some SEA weedy rice strains.

To examine whether the genetic diversity of SEA weedy rice is influenced by contemporary wild rice hybridization, we calculated the proportion of heterozygous genotype calls across the genome in each individual (i.e., the proportion of all variant sites within an individual that are heterozygous). Weedy rice, like cultivated rice, is predominantly self-fertilizing and is typically highly homozygous; therefore, an elevated proportion of heterozygous SNPs would be consistent with recent hybridization in an individual's ancestry. In line with this prediction, several of the SEA weedy rice accessions showed substantially elevated proportions of heterozygous SNPs in comparison to all other weedy and cultivated rice accessions (Fig. 2c). Considered together with the above phylogenetic, ancestry and diversity analyses, these results strongly suggest that SEA weedy rice has emerged through—and is continuing to evolve with—varying degrees of genetic contributions from wild rice.

### Wild rice introgression across the weedy rice genome
Weedy rice is characterized by some wild-like adaptive traits that could be directly derived from wild rice in locations where wild rice introgression occurs (e.g., seed shattering, seed dormancy, abiotic and biotic stress tolerance)[38]. In contrast, other traits contributed by wild rice could be maladaptive in the agricultural setting (e.g., prostrate growth, perenniality, irregular seed production). In the generations following a wild-to-weed hybridization event, one would therefore expect natural selection to favor the retention of some wild rice-derived genomic regions (i.e., those containing genes conferring weed-adaptive traits) and to favor the loss of others (those conferring maladaptive traits). To explicitly examine the fate of *O. rufipogon* variants across the genome in wild-introgressed weeds, we performed

genome scans to identify chromosomal regions that were differentially crop-like or wild-like based on SNPs characteristic of the two groups. In comparison to *indica*-like SEA weedy rice and Chinese weedy rice, the wild-like Thai weeds were characterized by large wild-like genomic blocks along the 12 chromosomes (Supplementary Fig. 5–8). Notably, these blocks included a number of rice domestication and improvement genes where the wild rice allele would be expected to be favored in weed populations; for example, within chromosome 4, wild-like genomic regions in these accessions encompassed domestication genes that control seed shattering (*sh4*), panicle architecture (*OsLG1*), awn length (which affects seed dispersal) (*An-1* and *LABA1*), and hull pigmentation (which affects grain conspicuousness to granivores) (*Bh4* and *Phr1*) (Supplementary Figs. 5–8 and Supplementary Data 6).

Because the chromosome-scale genome scans lacked resolution for identifying putatively-adaptive wild rice alleles in weed genotypes, we also surveyed allelic variation at specific known domestication genes for seed shattering, plant architecture and other traits. Modern crop cultivars phenotypically exhibit a domestication syndrome that is the outcome of a sequence of selective pressures, from early facultative encouragement of exploited wild species, to deliberate artificial selection during crop domestication and later varietal improvement. For the temperate weedy rice samples analyzed in the present study, both US weed strains (US-SH and US-BHA) and the Chinese weeds carry the crop-like alleles of three domestication genes that were early targets of selection: *PROG1* (erect plant architecture), *sh4* (reduced seed shattering) and *OsLG1* (closed panicle architecture) (Supplementary Data 6). This observation, together with extensive prior evidence[28,30], supports the de-domestication origin of these weed accessions. In contrast, while both the Thai and Malaysian weeds were fixed for the crop-like allele at *PROG1*, wild alleles of the *sh4* and *OsLG1* genes were identified in some of these SEA weedy rice accessions. The domestication allele of *PROG1* is likely under strong selection in weedy rice worldwide, which typically shows the erect growth that facilitates crypsis and competitive growth in crop fields. For the shattering genes, however, the presence of wild rice alleles in SEA weedy rice confirms the above observations that adaptive introgression from wild rice has contributed to the evolution of weediness traits in these strains. Interestingly, this pattern differs from most weedy rice populations worldwide, which, like the US and Chinese weeds, appear to have re-evolved shattering following selection for reduced shattering during domestication[28,30,48].

Besides *PROG1*, *sh4* and *OsLG1*, we also analyzed six widely selected domestication and two varietal-specific improvement genes. Most of the Malaysian weeds possesses a similar number of domestication alleles as the US-SH and Chinese weeds, whereas the Thai weedy accessions showed a greater presence of wild alleles across the six selected genes (Supplementary Data 6). This is again consistent with introgression of weed-adaptive wild rice alleles into the Thai weed strains.

### Identification of the candidate genes under selection
To test for genomic signatures of selection associated with weedy rice adaptation, we performed genome scans to identify low nucleotide diversity regions (LNDRs) that would be consistent with positive selective sweeps. The different weed samples varied in both the abundance and the chromosomal locations of LNDRs in comparisons to *indica* rice (Fig. 4). Malaysian and *indica*-like Thai weeds showed evidence of far more LNDRs than the wild-like Thai weeds; in addition, compared to the US weeds, where the top 5% of LNDRs were clustered within just a few chromosomal regions[30], the LNDRs identified in Malaysian and *indica*-like Thai weedy rice were scattered across the genome (Supplementary Data 7–9). As with LNDR scans, SweeD analyses[49] to detect signals of selective sweeps indicated that the majority of putative sweep regions differed among the different SEA weedy rice (Fig. 5 and Supplementary Data 10). Together, these

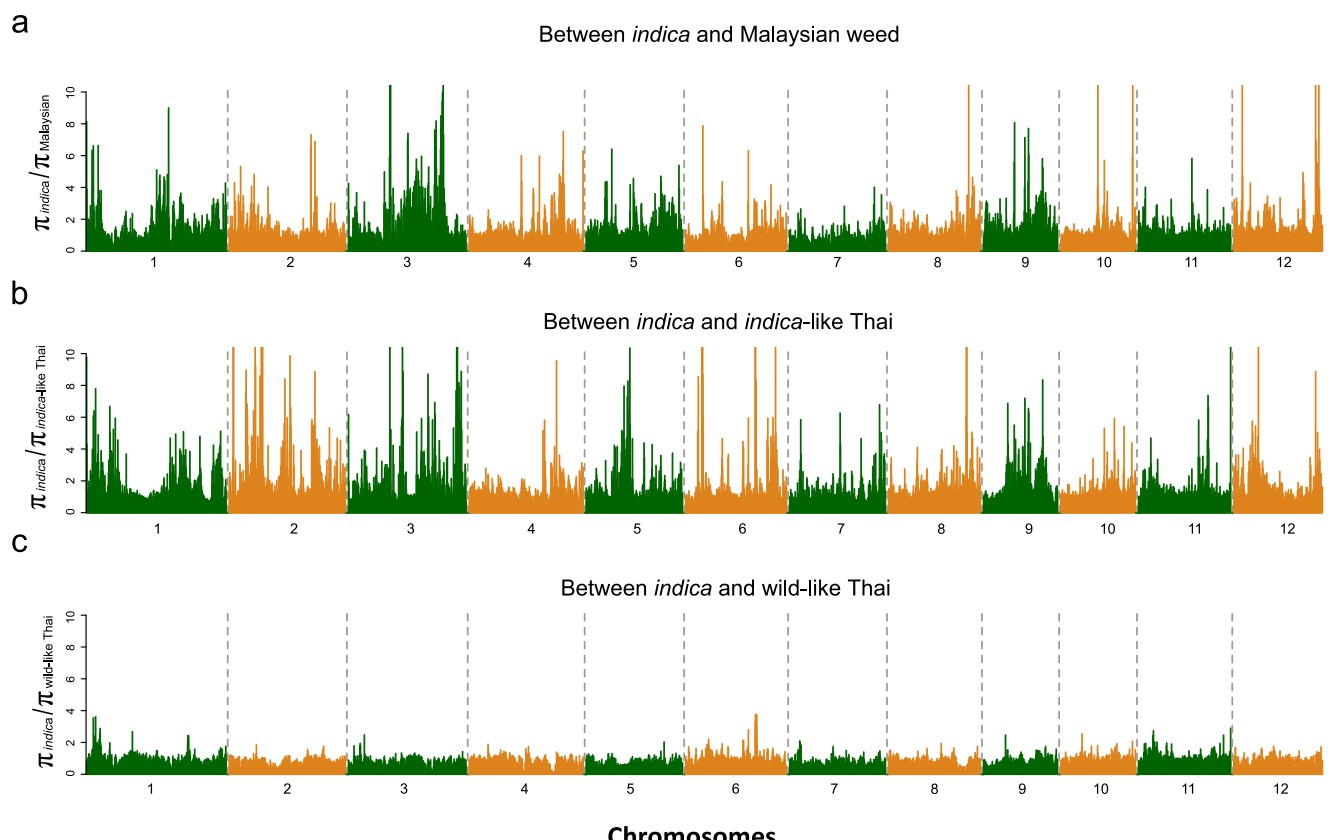

**Fig. 4 | Distribution of nucleotide diversity in the Malaysian, *indica*-like Thai and wild-like Thai weeds.** Numbers on the x-axis indicate the rice chromosome numbers. Ratio of nucleotide diversity between crop ancestor and weedy rice strains are shown on the y-axis for Malaysian (**a**), *indica*-like Thai (**b**), and wild-like Thai (**c**) weedy rice. Source data are provided as a Source Data file.

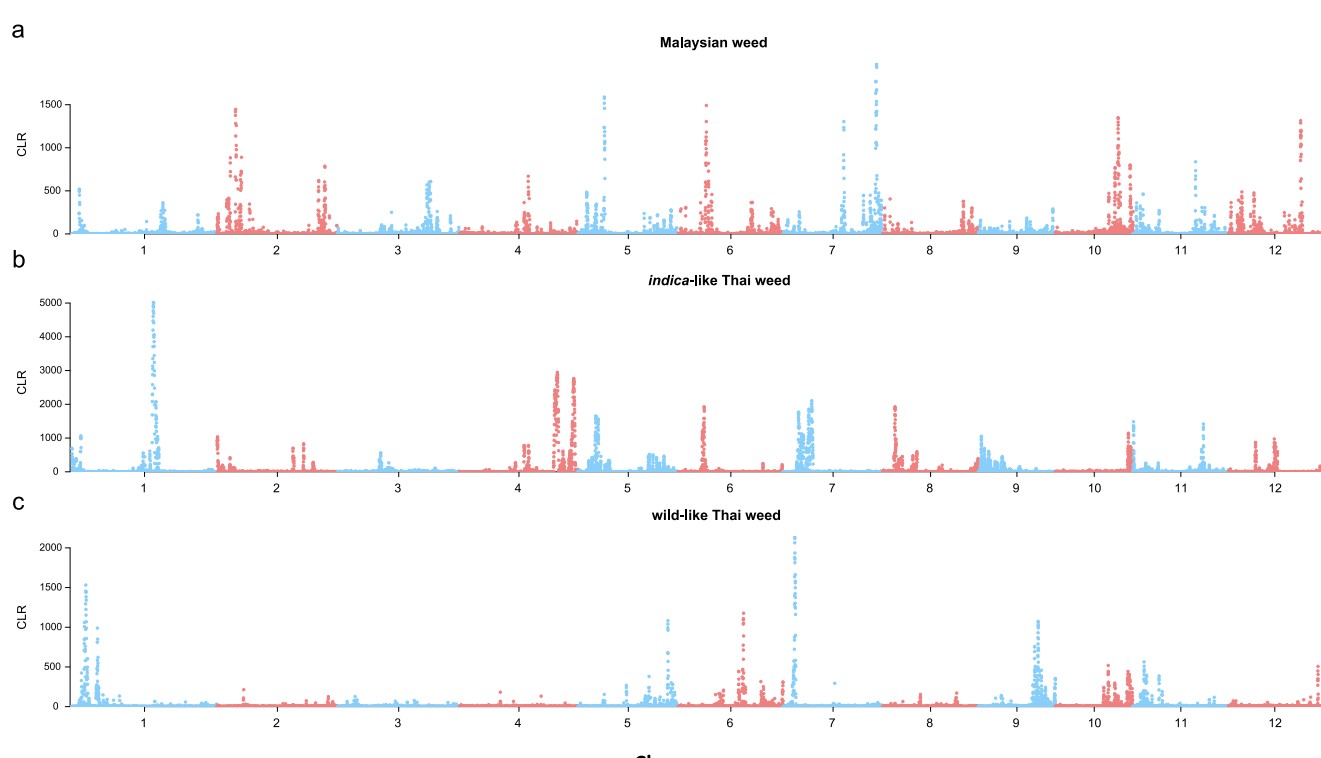

**Fig. 5 | Distributions of selective sweeps in the Malaysian, *indica*-like and wild-like Thai weeds.** Numbers on the x-axis indicate the rice chromosome numbers. Selection strength is indicated by composite likelihood ratio (CLR) values on the y-axis for Malaysian (**a**), *indica*-like Thai (**b**), and wild-like Thai (**c**) weedy rice. Source data are provided as a Source Data file.

analyses suggest that there have been distinct evolutionary trajectories in the selective processes shaping the different SEA weedy rice strains.

While the selection scans highlighted the genomic targets of selection that differed among the SEA weedy rice strains, we were also interested in examining the extent to which there was sharing of selection targets among them, and what candidate genes might occur in these shared regions. We identified a total of 2635 genes that are shared between at least two of these weedy strains (Supplementary Data 11), some of which could be of interest for further characterization. For example, two candidate genomic regions (chromosome 6: 5,300,000–5,400,000 and chromosome 12: 25,300,000–25,400,000) were shared among four weedy rice strains (US-BHA, US-SH, Malaysian and *indica*-like Thai weeds). However, reduction of nucleotide diversity in some of these genomic regions are likely to reflect neutral processes alone.

In Malaysian and Thai weeds, we further identified high genetic differentiation regions ($F_{ST}$) between the two weedy groups and their crop ancestors (Supplementary Data 7–9). In *indica*-like Thai weeds, for example, a candidate genomic region on chromosome 3 (26.0–26.1 Mb) contained several functionally important genes, such as lipid transfer protein (LTP) gene (LOC_Os03g46100 and LOC_Os03g46110) and F-box domain gene (LOC_Os03g46120 and LOC_Os03g46140). Plant LTP and F-box genes have been found to control many crucial cellular activities, including photomorphogenesis, floral meristem and floral organ identity determination, and response to abiotic stresses (i.e., drought, salt and cold)[50,51]. Likewise, a highly divergent genomic region (chromosome 10:11.4-11.6 Mb) was identified in Malaysian weeds. Candidate genes within this genomic region include two disease resistance genes (LOC_Os10g22290 and LOC_Os10g22300), both of which could be associated with adaptation of weedy strains.

## Discussion

While presenting particular challenges for weed control[52,53], weedy crop relatives can serve as useful models for studying the genomic underpinnings of rapid adaptation in the agricultural context[32,37]. A paradox of weedy rice adaptation in many world regions is its ability to so successfully exploit the rice agroecosystem despite its genetically depauperate background—as its genome carries both the 'cost of domestication' from its cultivated rice ancestor[4,9] and the bottlenecks and selective pressures imposed during feralization[28,30]. Here we have found that where weedy rice co-occurs with the rice wild progenitor (*O. rufipogon*), the porous species boundary has facilitated the process of weed adaptation.

Previous work, relying on morphology, specific genomic regions, and/or limited sample sizes, has suggested that the genetic composition of SEA weedy rice has been influenced by co-occurring wild rice as well as by gene flow from modern elite cultivars[33–35,38,39]. However, the extent of regional variation among SEA weedy rice strains, as well as the molecular mechanisms underlying the evolution of weediness traits, has remained largely unexplored. The present study used whole genome sequence analyses of Thai and Malaysian weedy rice compared to previously-published weedy rice genomes from temperate regions[28,30,40] to make inferences on the origin of SEA weedy rice populations and the role of wild rice introgression in their evolution. Our phylogenetic and population genomic analyses confirm that these weedy rice strains have evolved in part through wild rice introgression, including through wild-weed hybridization on a contemporary time scale. Moreover, in contrast to weedy rice populations from temperate regions where wild rice is absent, SEA weedy rice appears to have acquired weed-adaptive alleles in part through adaptive introgression from co-occurring wild rice, including for traits such as seed shattering and dispersal. For the different regions of SEA that were sampled, previous analyses of wild rice have indicated that Thai populations of the crop progenitor species are generally *indica*-like, while their counterparts in Malaysia are more *aus*-like[44,54]. Based on this information, we hypothesize that the Malaysian and Thai weedy rice strains may have obtained their wild-like alleles from *aus*-like and *indica*-like wild rice populations, respectively.

The evolution of weed-associated traits (i.e., seed shattering, seed dormancy, and rapid growth rate) is critical to the success of weedy rice strains for their ability to survive and persist in rice fields[20]. In the case of US weeds, we previously demonstrated that the majority of these weediness traits emerged through evolution of domesticated rice genomes in a process of de-domestication[29,30]. Similar phenomena were also found in weedy rice from other world regions, where standing variation and novel mutations together have resulted in the repeated occurrences of the weediness traits[28,31,40,48]. The results of the present analysis, which indicate that wild rice introgression can also play a role in this process, are broadly consistent with other studies which suggest multiple evolutionary pathways to weediness[28,36,38,39,55].

As genome sequence-based analyses of weedy rice populations from around the world have progressed over the last decade, it is becoming increasingly evident that weediness can evolve repeatedly, rapidly, and through diverse genetic mechanisms[28,30,31,48]. While de-domestication directly from crop ancestors is the most common mechanism, our genome sequence analyses of SEA weedy rice make it clear that genetic contributions from reproductively compatible wild *Oryzas* can also play a major role in this process. Moreover, while our genome scans for selection support the view that there are few genetic constraints on the mechanisms by which weediness emerges, we do nonetheless find a few shared targets among independently-evolved weed strains, including regions with genes controlling plant growth and development. However, evidence supporting the claims of our study are based on the available genome sequences of representative SEA weedy rice. With the advance of genome sequencing and other biotechnologies, further studies, based on a large sample size of pangenomes at multidimensional levels, together with gene editing for functional confirmation, will likely provide more comprehensive insights into the adaptive evolution of weedy rice. Given the major global economic impact of crop losses caused by this agricultural weed, further exploration of weediness related-genes could be useful for identifying genetic "Achilles heels" that could be exploited in future weed mitigation efforts.

## Methods

### Plant materials and sequencing

A total of 217 Asian and 10 African rice accessions were used in this study (Supplementary Data 1). Raw sequence data of the 183 Asian and 10 African rice accessions were downloaded from GenBank according to published references[30,40,42]. Additional details on the selection of previously sequenced rice accessions are provided in our previous study[30]. Newly sequenced genomes for the present study included one Malaysian wild rice accession, two elite Malaysian *indica* cultivars, and 31 SEA (18 Thai and 13 Malaysian) weedy rice accessions (Supplementary Data 1). In order to maximize the representation of genetic diversity in the sampled SEA weedy rice, these tropical Asian weedy accessions were selected based on their genetic background and morphological traits, as determined in our previous studies[29,34,35,37–39,56]. Genomic DNA was extracted from the mature leaves for each accession using DNAeasy Plant Mini Kits (QIAGEN, Germany) following the manufacturer's instructions. DNA libraries were constructed by Novogene (Tianjin, China), and paired-end sequencing was carried out on the Illumina HiSeq 2000 platform.

### Sequence mapping and variant genotyping

Data analyses of the 217 Asian and 10 African rice accessions were performed independently using the same pipeline. Clean reads were mapped onto the Nipponbare reference genome (*temperate japonica*, MSU 6.0 version, http://rice.plantbiology.msu.edu) using BWA[57] with

the parameter "bwa aln -n 0.05". Raw variants (insertions and deletions (INDELs) and SNPs) were then realigned with the Genome Analysis Toolkit (GATK) IndelRealigner version 2.6[58]. Genotype calling of the realigned assemblies was performed using SAMtools[59], with the parameter set as "mpileup -DSugf" and "bcftools view -Ncvg". All raw variants of the 217 Asian rice accessions were combined as a single variant call format (VCF) matrix. Genome sequences of the 10 African rice accessions were also mapped onto the Nipponbare reference genome (MSU 6.0 version) using the same pipeline. The two VCF matrices that were generated from Asian and African rice accessions were combined as an integrated dataset using VCFtools[60]. In our previous study, three different criteria were employed to eliminate systematic bias from the rice assemblies[30]; here we used the criterion "mapping quality ≥ 10 and read depth ≥ 1" to perform the raw variants filtering. Only reliable variants were then used for subsequent population genetic analyses.

## Phylogenetic and population genetic analyses

To infer phylogenetic relationships among Asian wild, weedy and cultivated rice accessions, we converted the VCF matrix into FASTA file using Perl scripts. Only SNPs that were homozygous and without missing data in any of the 217 rice accessions were applied to construct a Neighbor-Joining (NJ) tree using MEGA7[43] with the maximum composite likelihood substitution model. The resulting NJ tree was further rooted using wild rice Or-IIIa as the outgroup based on previous studies[30,42]. In addition, we also reconstructed NJ tree using the integrated SNP dataset of the Asian and African rice accessions with the same pipeline (Supplementary Fig. 9). Given that only 1,572,123 (21.32%) of the 7,372,500 variants that were identified in the 10 African rice accessions were shared with the 217 Asian rice accessions, subsequent population genomic inferences (i.e., nucleotide diversity and genetic differentiation) were performed for the 217 Asian rice accessions only. Maximum likelihood estimation of individual ancestries was performed using the program ADMIXTURE[61]. Variants (INDELs and SNPs) with missing data in any of the 217 rice accessions were excluded from the dataset. The ancestry populations were estimated with $K$ values from two to ten, and the lowest cross-validation error value was selected as the best ancestry populations (Supplementary Table 1). Principal component analysis (PCA) was performed using the same SNP dataset with program PLINK[62]. Gene flow between the weedy rice strains and their crop ancestors were estimated using Treemix[45]. Only the top two gene flow events were plotted within the topologies with colored arrows.

## Nucleotide variation distributions and selection scans

To assess the relatedness of the weedy rice strains to cultivated and wild rice, proportions of wild- and crop-specific private SNPs in the weedy rice genome were calculated according to our previous study[30]. Distribution patterns of these private SNPs were presented by plotting the log value of the ratio between the crop- and wild-specific private SNPs using R scripts. Genome-wide nucleotide diversity ($\pi$ and $\theta_W$) was calculated using VCFtools[60] for non-overlapping 100-kb windows across the genome. The reduction of nucleotide diversity between weedy stains and their crop ancestors ($\pi_{crop}/\pi_{weed}$) was estimated for non-overlapping windows across the genome using Perl scripts and the top 5% windows were treated as the candidates under selection during the adaptive evolution process. To further identify the genomic regions that have undergone selective sweep, the program SweeD[49] was applied to scan the genomes of these weedy rice stains at total and subgroup levels, respectively. Each chromosome was divided into 2,000 non-overlapping windows and signals of selective sweep were identified by the composite likelihood ratio (CLR) statistic[63]. As common weediness traits (i.e., seed shattering and dormancy) are shared among the weedy strains, we therefore identified the overlapped genomic regions by comparing the low nucleotide diversity windows of these weedy rice strains.

## Reporting summary

Further information on research design is available in the Nature Portfolio Reporting Summary linked to this article.

## Data availability

The raw total sequence reads have been deposited into Genome Warehouse in National Genomics Data Center, Chinese Academy of Sciences/China National Center for Bioinformation under the project number PRJCA016178. Source data are provided with this paper.

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

## Acknowledgements

We thank Olsen lab members for their constructive comments on drafts of this manuscript. This work was funded by the Malaysian Ministry of Higher Education (FRGS/1/2020/STG01/MUSM/02/4) and MUM-ASEAN Sustainable Development Research Grant (ASEAN-2019-01-SCI) to BKS, the NSF Plant Genome Research Program (IOS-1947609) to KMO, and National Science Foundation of China (31970235) to LFL.

## Author contributions

K.M.O., B.K.S., T.P. and L.F.L. conceived and designed this project. K.M.O., L.F.L., B.K.S., M.J.W. and T.P. wrote this manuscript. L.F.L., Y.L. Li, M.R.L., Y.L. Lau, S.J.Y., S.J., M.J.W., B.R. and Y.H. analyzed the data.

## Competing interests

The authors declare no competing interests.
