## [Peer Review File · Nature Communications]

Porous borders at the wild-crop interface promote weed adaptation in Southeast AsiaReviewers' Comments:

Reviewer #1:

Remarks to the Author:

This is a well-written manuscript that investigates origins of weedy rice that represents a significant problem for rice cultivation worldwide. This is an area of active scientific research and several papers of the last 10-15 years brought a good amount of information about the origins of weedy rice. It is now clear that weedy rice emerged multiple times via alternative pathways, even within the United States cultivation zone.

This article focuses on the question whether wild rice is involved, through hybridization with cultivated rice, in the emergence of weeds in southeast Asia, where wild and cultivated rice are sympatric. This has been long suspected, and to some extent also demonstrated before (<https://doi.org/10.1111/j.1365-294X.2007.03489.x>). The added value of this manuscript is that it investigates this question on a genome-wide scale, generating 34 new reference-based assemblies of weedy rice. The study also aims to identify candidate genes that are under selection in weedy rice and could represent an 'Achilles heel' that could help to fight the weeds. This is a very good justification of this research, nonetheless, I have a feeling that this is exactly where the present paper could provide a more in-depth analysis and more concrete information.

In particular, the authors identified 2,635 genes that have reduced nucleotide diversity in at least two of the weedy strains. Since low nucleotide diversity can be caused by neutral processes (founder effect, drift), a portion of these genes were probably not under selection and the low-diversity is shared by chance. I think the authors are aware of that, and therefore comment very briefly only on a couple of genomic regions shared among four weedy strains (lines 266-269). This is where more analyses are desired. It is not clear whether these four weed strains also have identical/similar haplotypes in these genomic regions, or the 'sharing' is limited to low diversity of various haplotypes. Perhaps these genomic scans can be combined with pairwise *Fst* scans to identify regions of fixation in weeds. Also, do the genes of these regions contain particular nonsynonymous substitutions that could be the target of selection (and the Achilles heel of weeds)? Furthermore, I might have missed it, but I did not find an integration of the SweeD result and the LNDR scans. A combination of high CLR and low diversity could narrow down the candidate genes. Perhaps a combination of the three metrics, CLR, diversity reduction and *Fst* could be the most informative.

I have a technical question about the SweeD scan. Personally, I have never managed to obtain proper results on crop species, and I think this is due to a bug in the program that was reported in this thread <https://groups.google.com/g/omegaplus/c/17Zc80PFz9g>. Basically, it seems SweeD does not report correct likelihoods for folded site-frequency spectra, which is when ancestral and derived states are not known. As I think this is also the case of your data, could you provide more details on how SweeD was run?

My other technical point is about the presented NJ tree. I really think that a PCA would be more appropriate to graphically represent the data. Cladistic concepts were not intended for population studies, especially in this case when hybridizations between groups are suspected. Indeed, the low bootstraps could indicate this problem, but that just means that the tree is very likely misleading. A minor point: lines 134-135 - the terminology is incorrect. If the weeds and wild rice form one clade, the group is paraphyletic, not monophyletic. In any case, if the weeds are hybrids, the terms monophyly or paraphyly do not apply. Also, the trees are presented as rooted, but there is no outgroup.

Other points:

line 97: Is it correct to say that temperate weeds have evolved solely through de-domestication? Can hybridization-derived weeds be imported into temperate regions via contaminated seeds? I think that was reported here <https://doi.org/10.1111/j.1365-294X.2007.03489.x> and elsewhere.

Lines 226-230: You say that the fact that the temperate weeds carry crop-like PROG1, OsLG1 and sh4 indicates that they originated via de-domestication. However, even weeds that originate through crop-wild hybridization could have crop-like haplotypes at those loci, simply by chance. Can the de-domestication of the temperate weeds be confirmed by direct evidence? The abstract (line 39) says that 'Thai and Malaysian weeds have acquired weed-associated traits through both de novo evolution and adaptive introgression', but I don't think you present direct evidence of de novo evolution. Is de novo evolution only implied, based on crop-like genome-wide ancestry?

line 162: that hypothesis has been confirmed by more recent studies (e.g. <https://doi.org/10.1093/gbe/evz039>)

Calculation of Pi from a VCF file is not theoretically proper, because it ignores invariant sites (see <https://doi.org/10.1111/1755-0998.13326>). Perhaps calculating the crop/weed ratio from Theta would be more appropriate.

Peter Civan, 04 May 2023

Reviewer #2:

Remarks to the Author:

This manuscript provides useful evidence that the adaption of weedy rice results from significant introgression of genes from wild rice. This is an issue that is important for understanding and managing weedy rice. It also has implications for the conservation of diversity in wild rice confirming gene flow between domesticated and wild rice populations. The data set is small by modern standards and based only on short read mapping. It would be preferable to see data on de novo genomes of some of the accession rather than just re-sequencing. Current technology makes this feasible. However, the analysis methods are all appropriate and make a compelling case. The potential for further work to identify the genetic mechanisms of weediness is of value.

Reviewer #3:

Remarks to the Author:

In the present manuscript, the authors have made an effort to reveal how the emergence of weedy rice in tropical Asian regions may be strongly influenced by hybridization with wild species. While I believe that the topic of the manuscript is valuable and helps us understand the evolution of weediness in weedy-type crops, I think the manuscript still has some significant limitations.

Firstly, I agree with the main conclusion that "wild rice introgression confers adaptability to Southeast Asian weedy rice," which has been reported many years ago. Unfortunately, this manuscript does not present any new findings or mechanisms for updating the evolution of Southeast Asian weedy rice.

Secondly, wild rice has good outcrossing habits, and gene exchange with neighboring rice is destined to happen. Although the authors succeeded in finding genomic evidence for a few cases by NGS, it cannot be regarded as a new rule or theory.

Lastly, I am concerned about whether such a small sample size (13 Malaysian and 18 Thai strains) is sufficient to reflect the objective diversity and population genomic information for Southeast Asian weedy rice. Performing population genetics analysis with this sample size would result in significant randomness. I believe that the results would change significantly if additional samples were added or some samples were replaced. Therefore, it is impossible to convincingly determine the genetic basis of weediness in weedy rice if the selection signal changes leading to changes in candidate genes

accordingly.

Reviewer #4:

Remarks to the Author:

This is a good paper that can potentially be a much better one. Most readers would find reasonably rigorous framing of the project, expertly execution and decent analyses of the data. Hence, it is an excellent fit for Nat Comm which has published many papers in this vein.

What then is the criticism? A reader cannot find anything beyond rice. Unless a biologist has some dealings with the rice researchers and rice literature, the paper would be consigned to a rice journal. The authors are so *Oryza*-centric, they seem to be writing the paper exclusively for agriculturists.

Can it be a much better paper, in concept? Yes, if done right, it can even be a great paper. How? The authors should expand the horizon into speciation, especially in the areas of gene flow and hybridization. It is not my job to inform the authors on such a common and important topic. But, for a start, there is a special issue on speciation and hybridization in NSR only recently (see below) which sends copies of this special issue to speciationists worldwide.

National Science Review 2022 Vol 9, Issue 12.

In this issue, the authors will find many topics in speciation that overlap theirs. For example, where in the process of evolution does gene flow between taxa cease? Are cultivated and wild rice sufficiently divergent that introgression takes on a different character than geographical populations? Also, parallel speciation has been discussed but dismissed as "nonsensical". I am sympathetic to the critics of parallel speciation as two divergent processes are unlikely to proceed in a similar fashion. The authors find a possible mechanism for parallel speciation - via hybridization between two parental species.

Unless the authors can expand the scope, this paper would be deemed too narrowly focused by most evolutionists. The expansion is conceptual, rather than experimental. So, it should be feasible during revisions but it will take a lot of reading, talking and, most of all, thinking about the essence of taxa divergence in general. Rice, cultivated, wild or weedy, is a window into that process.

I have other complaints about the presentation which is straightforward but template-derived. It is a bit too predictable. For example, the abstract says more or less that weedy rice is mainly cultivated rice with some introgressions from the wild relatives. What else can it be?

Below is an example of a similar story. Despite its simplicity, much like the current story, this Plos paper appeared in New York Times and many other media. I believe that the weedy rice story can be made into an NYT report. In short, the authors need to rewrite the piece to intrigue or even excite a wider audience in evolution and in agriculture.

He Z, Zhai W, Wen H, Tang T, Wang Y, et al. (2011) Two Evolutionary Histories in the Genome of Rice: the Roles of Domestication Genes. PLoS Genet 7(6): e1002100. doi:10.1371/journal.pgen.1002100

REVIEWER COMMENTS

Reviewer #1 (Remarks to the Author):

This is a well-written manuscript that investigates origins of weedy rice that represents a significant problem for rice cultivation worldwide. This is an area of active scientific research and several papers of the last 10-15 years brought a good amount of information about the origins of weedy rice. It is now clear that weedy rice emerged multiple times via alternative pathways, even within the United States cultivation zone.

Reply: We appreciate this positive comment and the detailed comments/suggestions the reviewer has provided below.

This article focuses on the question whether wild rice is involved, through hybridization with cultivated rice, in the emergence of weeds in southeast Asia, where wild and cultivated rice are sympatric. This has been long suspected, and to some extent also demonstrated before (<https://doi.org/10.1111/j.1365-294X.2007.03489.x>). The added value of this manuscript is that it investigates this question on a genome-wide scale, generating 34 new reference-based assemblies of weedy rice. The study also aims to identify candidate genes that are under selection in weedy rice and could represent an 'Achilles heel' that could help to fight the weeds. This is a very good justification of this research, nonetheless, I have a feeling that this is exactly where the present paper could provide a more in-depth analysis and more concrete information.

Reply: We performed data analyses accordingly. Please see our point-by-point response below.

In particular, the authors identified 2,635 genes that have reduced nucleotide diversity in at least two of the weedy strains. Since low nucleotide diversity can be caused by neutral processes (founder effect, drift), a portion of these genes were probably not under selection and the low-diversity is shared by chance. I think the authors are aware of that, and therefore comment very briefly only on a couple of genomic regions shared among four weedy strains (lines 266-269). This is where more analyses are desired. It is not clear whether these four weed strains also have identical/similar haplotypes in these genomic regions, or the 'sharing' is limited to low diversity of various haplotypes. Perhaps these genomic scans can be combined with pairwise F_{st} scans to identify regions of fixation in weeds. Also, do the genes of these regions contain particular nonsynonymous substitutions that could be the target of selection (and the Achilles heel of weeds)? Furthermore, I might have missed it, but I did not find an integration of the SweeD result and the LNDR scans. A combination of high CLR and low diversity could narrow down the candidate genes. Perhaps a combination of the three metrics, CLR, diversity reduction and F_{st} could be the most informative.

Reply: We agree with the reviewer that reduction of nucleotide diversity can be caused by neutral processes. As the reviewer suggested, a better way to narrow down the candidate genes is to employ the combination of CLR, diversity reduction and F_{st} . Given that the SweeD scan may cause some false positives (a point raised by the reviewer below), we therefore defined selective genomic regions as those that show overlap between low nucleotide diversity regions (LNDRs) (now based on both π and θ — see below) and high genetic differentiation regions (F_{st}). As shown in the new Supplementary Tables 7-9, using this approach we have identified some genomic regions that show reduction diversity and high differentiation between Malaysian/Thai weeds and their crop ancestors. These genomic regions are those we conclude

may potentially be under selection in weedy rice genome. We rephrased the related content. Please see details in the revised version of our manuscript (Page 10, Lines 301-315).

In addition, per the reviewer's suggestion, we also checked non-synonymous mutations of the six candidate genes (mentioned in Page 10, Lines 301-315). However, we did not identify non-synonymous mutations among these Southeast weeds, Malaysian *indica* and wild rice. With this reasoning, we did not provide detailed explanations in main text.

I have a technical question about the SweeD scan. Personally, I have never managed to obtain proper results on crop species, and I think this is due to a bug in the program that was reported in this thread <https://groups.google.com/g/omegaplus/c/17Zc80PFz9g>. Basically, it seems SweeD does not report correct likelihoods for folded site-frequency spectra, which is when ancestral and derived states are not known. As I think this is also the case of your data, could you provide more details on how SweeD was run?

Reply: We were able to get SweeD to run using the default parameters. However, we agree with the reviewer that while this program has been widely employed to estimate selective sweeps, it may not report correct likelihood values for folded site-frequency spectra. In our study, we intended to assess whether Thai and Malaysian weeds show distinct distribution patterns of the putative selective sweep regions (Figure 5). For our inferences on potential candidate genes, we followed the reviewer's recommendation in comment 1 above and focused on the regions of overlap in the LNDR and *Fst* analyses. We have retained the SweeD output to allow for a qualitative comparison.

My other technical point is about the presented NJ tree. I really think that a PCA would be more appropriate to graphically represent the data. Cladistic concepts were not intended for population studies, especially in this case when hybridizations between groups are suspected. Indeed, the low bootstraps could indicate this problem, but that just means that the tree is very likely misleading. A minor point: lines 134-135 - the terminology is incorrect. If the weeds and wild rice form one clade, the group is paraphyletic, not monophyletic. In any case, if the weeds are hybrids, the terms monophyly or paraphyly do not apply. Also, the trees are presented as rooted, but there is no outgroup.

Reply: Per this suggestion, we have performed a PCA for the same accessions, and we now present those results in the paper (see the new Supplementary Figure 2) (Page 6, Lines 160-168; Page 13, Lines 417-418). We also corrected the terminology within the text. All new changes are shown in blue (Page 6, Lines 163-164). Rather than removing the NJ tree from the paper, we prefer to keep both analyses, as we believe there is value in the comparison and more generally in presenting genetic distance trees with these *Oryza* data — given the deep phylogenetic divergence between the *indica* and *japonica* lineages of *O. sativa* (~400 kya), as well as additional divergence in *O. rufipogon/nivara*. Tree-like representations like ours have been commonly used in other studies of the Asian rice complex (*e.g.*, recent pangenome paper by Wu *et al.*, 2023, *Genome Biology*, <https://doi.org/10.1186/s13059-023-03017-5>). We take the reviewer's point that hybridization violates the assumption of a strictly bifurcating tree, but the deeper branches in our tree provide a useful graphic for visualizing the independent weedy rice origins. In this way the NJ tree is complementary to the PCA.

Regarding the rooting, the NJ tree is rooted based on the topologies generated in a previous study (Huang *et al.*, 2012, *Nature*). Huang *et al.* defined the wild rice as three groups (Or-I, Or-II and Or-III) and used the group Or-IIIa as the root (Fig. 2a in Huang *et al.*, 2012). As all genome sequences of the wild rice accessions used in our study were obtained from Huang *et al.* (2012), we then root the NJ tree using the wild rice

group Or-IIIa as in our previous study (Li *et al.*, 2017, *Nature Genetics*). Here we used the same strategy to root the NJ tree. To clarify this rooting methodology, we added the related content in the revised version of our manuscript (Page 13, Lines 411-412) and legend of Figure 1.

Other points:

line 97: Is it correct to say that temperate weeds have evolved solely through de-domestication? Can hybridization-derived weeds be imported into temperate regions via contaminated seeds? I think that was reported here <https://doi.org/10.1111/j.1365-294X.2007.03489.x> and elsewhere.

Reply: We agree with the reviewer that some temperate weeds could originate through introductions of strains that evolved elsewhere, and that these could potentially include strains that evolved through hybridization. As the reviewer correctly notes with the cited reference (Londo and Schaal, 2007), weedy rice in the USA is descended from strains that originally evolved in Asia. Subsequent to that 2007 study (based on SSRs), we determined via WGS analysis that US weedy rice evolved through de-domestication of Asian *indica* and *aus* cultivated varieties (see, *e.g.*, Li *et al.*, 2017, *Nat. Genet.*). Additional WGS analyses using a global sample of weedy rice further demonstrated that, outside of the USA, all or nearly all other temperate weedy rice strains have evolved directly from local rice (Qiu *et al.*, 2020). We are not aware of any evidence that temperate weedy rice has evolved through mechanisms other than de-domestication of temperate crop varieties. Nonetheless, we agree that “solely” may be a bit overconfident, and we have moderated the wording here (Page 4, Lines 92-95).

Lines 226-230: You say that the fact that the temperate weeds carry crop-like *PROG1*, *OsLG1* and *sh4* indicates that they originated via de-domestication. However, even weeds that originate through crop-wild hybridization could have crop-like haplotypes at those loci, simply by chance. Can the de-domestication of the temperate weeds be confirmed by direct evidence? The abstract (line 39) says that 'Thai and Malaysian weeds have acquired weed-associated traits through both *de novo* evolution and adaptive introgression', but I don't think you present direct evidence of *de novo* evolution. Is *de novo* evolution only implied, based on crop-like genome-wide ancestry?

Reply: In our previous study (Li *et al.*, 2017, *Nat. Genet.*), the de-domestication origin of US weed was inferred by the combination of domestication genes, relative divergence time and genome-wide nucleotide variation pattern. We agree with the reviewer that these lines of evidence are still not definitive, and we have rephrased the related content in the revision (Page 2, Lines 43-44; Page 9, Lines 257-262).

line 162: that hypothesis has been confirmed by more recent studies (e.g. <https://doi.org/10.1093/gbe/evz039>).

Reply: Thanks for pointing this out. We updated this text accordingly (Page 7, Lines 191-194).

Calculation of Pi from a VCF file is not theoretically proper, because it ignores invariant sites (see <https://doi.org/10.1111/1755-0998.13326>). Perhaps calculating the crop/weed ratio from Theta would be more appropriate.

Reply: We agree with the reviewer that calculation of π from variable sites only is theoretically problematic (although it does provide a useful way of comparatively quantifying genetic diversity across

the genome to identify LDNRs). As shown in the revised Supplementary Tables 7-9, we have now calculated both the π and θ values for each 100-kb non-overlap sliding window. Only those genomic regions that show decrease in nucleotide diversity in both the π and θ ratios are now defined as low nucleotide diversity regions.

Reviewer #2 (Remarks to the Author):

This manuscript provides useful evidence that the adaption of weedy rice results from significant introgression of genes from wild rice. This is an issue that is important for understanding and managing weedy rice. It also has implications for the conservation of diversity in wild rice confirming gene flow between domesticated and wild rice populations. The data set is small by modern standards and based only on short read mapping. It would be preferable to see data on *de novo* genomes of some of the accession rather than just re-sequencing. Current technology makes this feasible. However, the analysis methods are all appropriate and make a compelling case. The potential for further work to identify the genetic mechanisms of weediness is of value.

Reply: We appreciate this critical comment. Our key inferences in this study are based on comparing the SE Asian weeds to a global sample of rice accessions, providing a dataset of 217 diverse accessions. Like our samples, the vast majority of the previously published rice genomes were generated through short-read sequencing. While we agree with the reviewer that it would be ideal to have *de novo* genome assemblies for the entire dataset (*i.e.*, a pangenome comprising a global sample of wild, cultivated and weedy rice), having *de novo* genomes for our samples alone wouldn't do us much good for the comparative analyses that form the core of the study. We further note that there is now an extensive body of literature showing that population genomic inferences based on short read mapping can provide detailed and robust insights into weedy rice evolution.

Per the reviewer's key point, we now highlight the value of pangenome analysis as a "next step" in our Discussion, and we clarify the criteria for selecting accessions for the comparative dataset in the Methods. (Page 4, Lines 111-112; Page 12, Lines 373-375; Pages 12-13 385-388).

Reviewer #3 (Remarks to the Author):

In the present manuscript, the authors have made an effort to reveal how the emergence of weedy rice in tropical Asian regions may be strongly influenced by hybridization with wild species. While I believe that the topic of the manuscript is valuable and helps us understand the evolution of weediness in weedy-type crops, I think the manuscript still has some significant limitations.

Reply: We appreciate the reviewer's critique and comments below.

Firstly, I agree with the main conclusion that "wild rice introgression confers adaptability to Southeast Asian weedy rice," which has been reported many years ago. Unfortunately, this manuscript does not present any new findings or mechanisms for updating the evolution of Southeast Asian weedy rice.

Reply: As noted by Reviewer 1 above, the new genome sequence analyses in this study provide a valuable step forward. Evolutionary mechanisms underlying the origin and adaptation of Southeast Asian weedy rice have been proposed by some of our previous studies (*i.e.*, Song *et al.*, 2014, *Mol. Ecol.*; Qi *et al.*, 2015, *Mol. Ecol.*; Cui *et al.*, 2016, *G3*). However, the genome architecture and selective signatures of these hybrid-derived weedy strains remained unclear. Here our results indicate that these Southeast Asian weedy rice strains show high genome dynamics. Candidate genes under selection are functionally related to abiotic stress, disease resistance and development. Thus, our new study provides important genomic insight into evolution of Southeast Asian weedy rice via wild rice introgression.

Per the reviewer's comment, we have clarified the conceptual advance represented by our work in the revised version of the manuscript (Page 4, Lines 111-112).

Secondly, wild rice has good outcrossing habits, and gene exchange with neighboring rice is destined to happen. Although the authors succeeded in finding genomic evidence for a few cases by NGS, it cannot be regarded as a new rule or theory.

Reply: Weedy rice, like cultivated rice, is highly selfing. As such, introgression from outcrossing wild populations growing near rice fields is by no means a foregone conclusion. In the absence of selection for adaptive introgression (as we have documented here), we suspect that hybridization rates would in fact be very low. More generally, understanding how genetic introgression has contributed to adaptation and speciation has long been a central issue in evolutionary biology. Here we identified adaptive introgressions from wild rice to weedy rice, as well as the genetic basis underlying the selection. Our newly framed Introduction (Page 2, Lines 32-33; Page 3, Lines 52-79; Page 11, Lines 318-327) now highlights this conceptual framework.

Lastly, I am concerned about whether such a small sample size (13 Malaysian and 18 Thai strains) is sufficient to reflect the objective diversity and population genomic information for Southeast Asian weedy rice. Performing population genetics analysis with this sample size would result in significant randomness. I believe that the results would change significantly if additional samples were added or some samples were replaced. Therefore, it is impossible to convincingly determine the genetic basis of weediness in weedy rice if the selection signal changes leading to changes in candidate genes accordingly.

Reply: In our previous studies, we examined the origin of Southeast Asian weedy rice using large sample sets (*e.g.*, 133 weed accessions in Vigueira *et al.*, 2019, *Evol. Appl.*). In the present study, the weedy rice accessions were selected to represent Southeast Asian weedy rice based on their genetic background and morphological traits. Per the reviewer's comment, we have clarified the criteria for choosing the accessions in the revised version of our manuscript (Pages 12-13, Lines 385-388).

Reviewer #4 (Remarks to the Author):

This is a good paper that can potentially be a much better one. Most readers would find reasonably rigorous framing of the project, expertly execution and decent analyses of the data. Hence, it is an excellent fit for Nat Comm which has published many papers in this vein.

Reply: We appreciate the reviewer's positive overall assessment of the paper and its fit to *Nature Communications*.

What then is the criticism? A reader cannot find anything beyond rice. Unless a biologist has some dealings with the rice researchers and rice literature, the paper would be consigned to a rice journal. The authors are so *Oryza*-centric, they seem to be writing the paper exclusively for agriculturists.

Reply: The reviewer has raised an excellent point. As detailed in our responses below, the revision now frames the study in a broader conceptual framework that allows us to highlight its evolutionary implications beyond the rice system.

Can it be a much better paper, in concept? Yes, if done right, it can even be a great paper. How? The authors should expand the horizon into speciation, especially in the areas of gene flow and hybridization. It is not my job to inform the authors on such a common and important topic. But, for a start, there is a special issue on speciation and hybridization in NSR only recently (see below) which sends copies of this special issue to speciationists worldwide.

National Science Review 2022 Vol 9, Issue 12.

In this issue, the authors will find many topics in speciation that overlap theirs. For example, where in the process of evolution does gene flow between taxa cease? Are cultivated and wild rice sufficiently divergent that introgression takes on a different character than geographical populations? Also, parallel speciation has been discussed but dismissed as "nonsensical". I am sympathetic to the critics of parallel speciation as two divergent processes are unlikely to proceed in a similar fashion. The authors find a possible mechanism for parallel speciation - via hybridization between two parental species. Unless the authors can expand the scope, this paper would be deemed too narrowly focused by most evolutionists. The expansion is conceptual, rather than experimental. So, it should be feasible during revisions but it will take a lot of reading, talking and, most of all, thinking about the essence of taxa divergence in general. Rice, cultivated, wild or weedy, is a window into that process.

Reply: We thank the reviewer for pointing us to these themes relating to reproductive isolation/speciation and the impacts of secondary hybridization and gene flow. Looking into these topics in the context of our system, we realized that our study provides valuable new insights in two key areas: 1) the role of adaptive introgression as a mechanism that can overcome the 'cost of domestication' in agricultural weeds descended from domesticated ancestors (which are the norm for weedy rice — see next comment below); and 2) how for rice in particular, the lack of reproductive isolating barriers between the domesticated and wild species (which has previously been recognized as preventing complete speciation in this system — see, *e.g.*, Mi *et al.*, 2020) has had the unfortunate consequence of enhancing adaptation in a problematic agricultural weed – weedy rice.

The revised manuscript now presents these ideas as the opening of the Introduction (Page 3, Lines 52-79) and elsewhere, *e.g.*, sections *Abstract* (Page 2, Lines 32-33) and *Discussion* (Page 11, Lines 318-327). We also changed the title of this manuscript (Page 1, Lines 3-4).

I have other complaints about the presentation which is straightforward but template-derived. It is a bit too predictable. For example, the abstract says more or less that weedy rice is mainly cultivated rice with some introgressions from the wild relatives. What else can it be?

Reply: In fact, the vast majority of weedy rice strains worldwide are direct descendants of domesticated crop varieties. We now emphasize this fact in the revised text (Page 2, Lines 32-33; Page 3, Lines 52-73; Page 4, Lines 92-95; Page 11, Lines 318-327). For Asia in particular, our 2020 WGS analysis documented that a strikingly high proportion of contemporary weedy rice strains are direct descendants of “elite” hybrid cultivars that were widely introduced in the 1980s.

Below is an example of a similar story. Despite its simplicity, much like the current story, this Plos paper appeared in New York Times and many other media. I believe that the weedy rice story can be made into an NYT report. In short, the authors need to rewrite the piece to intrigue or even excite a wider audience in evolution and in agriculture.

He Z, Zhai W, Wen H, Tang T, Wang Y, et al. (2011) Two Evolutionary Histories in the Genome of Rice: the Roles of Domestication Genes. *PLoS Genet* 7(6): e1002100. doi:10.1371/journal.pgen.1002100

Reply: We again thank the reviewer for prodding us to think bigger (and beyond rice) in framing our study. We believe we have accomplished just that with the revised text, and that the paper is greatly improved as a result.

Reviewers' Comments:

Reviewer #1:

Remarks to the Author:

I am mostly satisfied with the authors' responses to my comments, and the corrections in the text. I accept all conclusions of the manuscript, but there are still some minor technical issues, that could be corrected to improve the overall quality.

I still feel a little bit uneasy with the NJ tree. I am aware that trees are often used to represent populations, despite populations violating cladistic assumptions. To some extent, I'm guilty of that in my own papers. However, here I am not sure what is the utility of the tree, given that *indica* is scattered all over the tree, and the justification for using Or-III accessions as an outgroup is weak. Now it occurred to me that outgroup information is also essential for Treemix, reported on Supplementary Fig. 3. Here, the outgroup is shown as 'wild rice', and it is unclear whether this is all sampled *O. rufipogon*, or just the Or-III accessions. Either way, it is strange to have an outgroup that is suspected to interbreed with ingroups, and I would expect that Treemix does not perform well in such situations (the migration events can, by definition, only involve ingroup branches). Indeed, the migration events are not inferred to come from wild rice (as reported), but rather from the internal branches of the weedy-cultivated clade, causing some confusion.

It is not too difficult to solve this problem. Sequencing data for an appropriate outgroup is available (*O. barthii*, *O. glaberrima*, perhaps *O. longistaminata*). These can be mapped onto the reference, and SNPs called at the exact positions that have been used for the analyses (without the need to re-build the entire VCF file). These data can be used in the NJ tree, and more importantly, in Treemix.

Lines 220-229:

I understand what was done here, and I agree it is a useful measure. However, there could be some confusion in terminology. What is usually understood under H_o (observed heterozygosity), is the proportion of heterozygous genotypes at a particular locus across a population sample. Here however, it means the proportion of heterozygous genotypes in a particular individual across all genotyped loci. I suggest this should not be called H_o , but instead just descriptively 'proportion of heterozygous genotype calls in an individual'.

Wouldn't it be interesting to add the US weeds to the Fig. 3?

I am still intrigued by the idea of the 'Achilles heel' of weeds. It is an attractive idea, but I think I am missing something here. The way I understand it is that if we find some universal genetic features of weeds, these could be subsequently used to combat the weeds. But how exactly would that work? We cannot control the genetic make-up of weeds, i.e. we cannot combat existing weeds by breeding different weeds. Or, are you thinking about changes in agricultural practices? It is hard for me to imagine what should be looked for. Is there an example from a different crop-weed system? Also, the conclusion on lines 291-295 should be emphasized in this context. I would put it bluntly - the weedy rice strains compared here have little in common on the genetic level, making a universal genome-informed mitigation strategy difficult/unlikely.

A final minor suggestion: 'SEA weeds' (10 occurrences in the main text) feels a bit odd. I know it's perfectly clear from the context, but still, I would replace it with 'SEA weedy rice'.

Peter Civan, 03 October 2023

Reviewer #2:

Remarks to the Author:

I am satisfied with the responses that the authors have provided.

Reviewer #3:

Remarks to the Author:

In the present manuscript, the authors have made an effort to reveal how the emergence of weedy rice in tropical Asian regions may be strongly influenced by hybridization with wild species. While I believe that the topic of the manuscript is valuable and helps us understand the evolution of weediness in weedy-type crops, I think the manuscript still has some significant limitations.

Reply: We appreciate the reviewer's critique and comments below.

Firstly, I agree with the main conclusion that "wild rice introgression confers adaptability to Southeast Asian weedy rice," which has been reported many years ago. Unfortunately, this manuscript does not present any new findings or mechanisms for updating the evolution of Southeast Asian weedy rice.

Reply: As noted by Reviewer 1 above, the new genome sequence analyses in this study provide a valuable step forward. Evolutionary mechanisms underlying the origin and adaptation of Southeast Asian weedy rice have been proposed by some of our previous studies (i.e., Song et al., 2014, Mol. Ecol.; Qi et al., 2015, Mol. Ecol.; Cui et al., 2016, G3). However, the genome architecture and selective signatures of these hybrid-derived weedy strains remained unclear. Here our results indicate that these Southeast Asian weedy rice strains show high genome dynamics. Candidate genes under selection are functionally related to abiotic stress, disease resistance and development. Thus, our new study provides important genomic insight into evolution of Southeast Asian weedy rice via wild rice introgression.

Per the reviewer's comment, we have clarified the conceptual advance represented by our work in the revised version of the manuscript (Page 4, Lines 111-112).

Feedback: Thank you for your reply. The genome architecture and selective signatures considering as the 'important genomic insight into evolution' mentioned by author are just my main concern. Because candidate genes derived from selection sweep and annotation, without any biological experimental evidence, can be uncertain and should not be used as a conclusion, especially with a small sample size in population genetics.

I very much look forward to the authors providing new data or evidence to refute your comment. However, I noticed that the author only added a brief sentence in the introduction in response to my comment, stating that "the genome architecture and selective signatures of tropical Asian weedy strains remain largely unclear."

Secondly, wild rice has good outcrossing habits, and gene exchange with neighboring rice is destined to happen. Although the authors succeeded in finding genomic evidence for a few cases by NGS, it cannot be regarded as a new rule or theory.

Reply: Weedy rice, like cultivated rice, is highly selfing. As such, introgression from outcrossing wild populations growing near rice fields is by no means a foregone conclusion. In the absence of selection for adaptive introgression (as we have documented here), we suspect that hybridization rates would in fact be very low. More generally, understanding how genetic introgression has contributed to

adaptation and speciation has long been a central issue in evolutionary biology. Here we identified adaptive introgressions from wild rice to weedy rice, as well as the genetic basis underlying the selection. Our newly framed Introduction (Page 2, Lines 32-33; Page 3, Lines 52-79; Page 11, Lines 318-327) now highlights this conceptual framework.

Feedback: Thank you for your reply. It is common knowledge among rice-related researchers that even temperate japonica rice, which has the lowest stigma-exposed rate among ecotypes, can have an outcrossing rate of up to 5%. Weedy rice, indica rice, and wild rice have much higher stigma-exposed rates. While "destined to happen" may be an extreme phrase (due to certain reproductive isolation), it is reasonable to say that the emergence of a new ecotype of weedy rice is "not unexpected" due to this relatively high outcrossing rate. The authors' evidence for key domestication genes, such as *sh4*, supporting adaptive introgression from wild rice to weedy rice, is important. I understand and agree with your view, however, this evidence may not be exciting or innovative enough for potential readers in evolutionary biology and crop research.

Regarding the newly framed Introduction, I agree that it effectively highlights the conceptual framework and increases the quality and value of the manuscript. However, I also agree that experimental and analytical evidence are better than explanations and citations to the research papers.

Lastly, I am concerned about whether such a small sample size (13 Malaysian and 18 Thai strains) is sufficient to reflect the objective diversity and population genomic information for Southeast Asian weedy rice. Performing population genetics analysis with this sample size would result in significant randomness. I believe that the results would change significantly if additional samples were added or some samples were replaced. Therefore, it is impossible to convincingly determine the genetic basis of weediness in weedy rice if the selection signal changes leading to changes in candidate genes accordingly.

Reply: In our previous studies, we examined the origin of Southeast Asian weedy rice using large sample sets (e.g., 133 weed accessions in Vigueira et al., 2019, *Evol. Appl.*). In the present study, the weedy rice accessions were selected to represent Southeast Asian weedy rice based on their genetic background and morphological traits. Per the reviewer's comment, we have clarified the criteria for choosing the accessions in the revised version of our manuscript (Pages 12-13, Lines 385-388).

Feedback: Thank you for your reply. I agree that the weedy rice accessions selected based on their genetic background and morphological traits can represent Southeast Asian weedy rice. However, the evasive answer still fails to address my concern effectively about whether the results would significantly change if additional samples were added or some samples were replaced. In population genetics, sample size determines accuracy to some extent based on the basic principles of statistics. While I agree with the representativeness of the sample, I have doubts about whether a small number of representative samples can reflect statistical accuracy enough to obtain general rules. Specifically, I am curious whether the selection sweeps for a small number of samples and a large number of samples overlap in this study. Whether or not there is an overlap is not an issue for me, but I regret that the author did not attempt to investigate this further.

Reviewer #4:

Remarks to the Author:

The revisions are satisfactory, congratulations.

REVIEWER COMMENTS

Reviewer #1 (Remarks to the Author):

I am mostly satisfied with the authors' responses to my comments, and the corrections in the text. I accept all conclusions of the manuscript, but there are still some minor technical issues, that could be corrected to improve the overall quality.

Reply: We appreciate the reviewer for all his previous comments and suggestions, which largely improved our manuscript. We also made changes according to the new comments and suggestions. Please see our point-by-point responses below.

I still feel a little bit uneasy with the NJ tree. I am aware that trees are often used to represent populations, despite populations violating cladistic assumptions. To some extent, I'm guilty of that in my own papers. However, here I am not sure what is the utility of the tree, given that *indica* is scattered all over the tree, and the justification for using Or-III accessions as an outgroup is weak. Now it occurred to me that outgroup information is also essential for Treemix, reported on Supplementary Fig. 3. Here, the outgroup is shown as 'wild rice', and it is unclear whether this is all sampled *O. rufipogon*, or just the Or-III accessions. Either way, it is strange to have an outgroup that is suspected to interbreed with ingroups, and I would expect that Treemix does not perform well in such situations (the migration events can, by definition, only involve ingroup branches). Indeed, the migration events are not inferred to come from wild rice (as reported), but rather from the internal branches of the weedy-cultivated clade, causing some confusion. It is not too difficult to solve this problem. Sequencing data for an appropriate outgroup is available (*O. barthii*, *O. glaberrima*, perhaps *O. longistaminata*). These can be mapped onto the reference, and SNPs called at the exact positions that have been used for the analyses (without the need to re-build the entire VCF file). These data can be used in the NJ tree, and more importantly, in Treemix.

Reply: We appreciate this comment! Accordingly, we downloaded genome sequences of five *O. barthii* and five *O. glaberrima* accessions from NCBI. Then, we performed SNP calling for the 10 African rice accessions using the same reference genome (*O. sativa* ssp. *japonica*, MSU version 6.0). In our previous study, we performed joint-calling for all 217 accessions from BAM files (see in Li *et al.*, 2017, *Nature Genetics*). With this reasoning, we also performed joint-calling for the 10 African rice accessions. Then, we combined the two SNP datasets using the program VCFtools. Given that African rice (285 Mb genome size for *O. glaberrima* and 308 Mb for *O. barthii*) and Asian rice (374 Mb for *O. sativa* ssp. *japonica*) show different genome features and are phylogenetically distinct to each other (Stein *et al.*, 2018, *Nat. Genet.*), we only included these variants that could be identified in common in the two independent SNP datasets in subsequent genomic population inferences.

Firstly, based on the 227 accessions (217 Asian rice and 10 African rice) of the combined SNP dataset, we reconstructed the NJ tree using the ten African rice accessions as the outgroup. As shown in the new Supplementary Figure 6, tree topologies are slightly different from the previous one (see in the Figure 1 and Supplementary 1, not including the 10 African rice accessions), particularly in the clade that contains the cultivated rice varieties (*japonica* and *aromatic*) and wild rice. For example, these wild Asian rice accessions did not form a monophyletic clade in the new NJ tree, while high bootstrap support values were obtained. It should be noted that this new NJ tree (Supplementary Figure 6) was built based on only the SNPs that could be identified in common across the newly added African rice accessions and our existing accessions (1,572,123 SNPs, only 135,670 of which are homozygous SNPs). In contrast, our previous NJ tree (Supplementary Figure 1) was built based on 1,455,426 homozygous SNPs.

Secondly, we also performed a PCA using the combined SNP dataset (including the 10 African rice accessions). Yet, whereas our previous PCA (see in Supplementary Figure 2) clearly shows that these weedy rice strains grouped with their crop ancestors, the new PCA results (see below) did not separate these weedy rice strains and the crop subspecies. In particular, the 10 African rice accessions mixed with Asian rice accessions. A similar pattern was also observed in the Treemix analyses where a strange phylogenetic tree was generated by this program (see below). We think that this can be explained by the low number of high-quality SNPs that we could identify across this phylogenetically broader group of *Oryza* species. In particular, all these SNPs were identified based on the Asian rice reference genome.

Note: PCA of the 217 Asian and 10 African rice accessions based on common SNPs.

Note: Phylogenetic topologies of the Asian and African rice groups based on common SNPs.

Because of the ambiguous results from these new analyses, we only updated the new NJ tree (the new Supplementary Figure 6). We also added the related content in the revised version of our manuscript (Page 13, Lines 405-416; Page 14, Lines 428-434).

Lines 220-229:

I understand what was done here, and I agree it is a useful measure. However, there could be some confusion in terminology. What is usually understood under H_o (observed heterozygosity), is the proportion of heterozygous genotypes at a particular locus across a population sample. Here however, it means the proportion of heterozygous genotypes in a particular individual across all genotyped loci. I suggest this should not be called H_o , but instead just descriptively 'proportion of heterozygous genotype calls in an individual'.

Reply: Thank you for pointing out this important wording clarification. Accordingly, we changed it to "the proportion of heterozygous genotype calls across the genome in each individual" (Page 8, Lines 229-239).

Wouldn't it be interesting to add the US weeds to the Fig. 3?

Reply: Per this comment, US weeds were included into the new Figure 3. We also added the related content based on the new Figure 3 (Pages 7-8, Lines 217-226).

I am still intrigued by the idea of the 'Achilles heel' of weeds. It is an attractive idea, but I think I am missing something here. The way I understand it is that if we find some universal genetic features of weeds, these could be subsequently used to combat the weeds. But how exactly would that work? We cannot control

the genetic make-up of weeds, *i.e.*, we cannot combat existing weeds by breeding different weeds. Or, are you thinking about changes in agricultural practices? It is hard for me to imagine what should be looked for. Is there an example from a different crop-weed system? Also, the conclusion on lines 291-295 should be emphasized in this context. I would put it bluntly - the weedy rice strains compared here have little in common on the genetic level, making a universal genome-informed mitigation strategy difficult/unlikely.

Reply: We agree with the reviewer that it is very hard to find common genetic mechanisms underlying the formation of weediness traits. However, genome sequence-based analyses of worldwide weedy rice strains have indeed revealed genetic basis of weediness traits over the last decade. Right now, we are still on the way to find the genetic "Achilles heel". As we mentioned, further studies, such as pangenomes of worldwide rice accessions, might provide a more efficient avenue to resolve this problem. Per this comment, we revised the related content (Page 12, Lines 381-386).

A final minor suggestion: 'SEA weeds' (10 occurrences in the main text) feels a bit odd. I know it's perfectly clear from the context, but still, I would replace it with 'SEA weedy rice'.

Reply: We made changes accordingly.

Reviewer #2 (Remarks to the Author):

I am satisfied with the responses that the authors have provided.

Reply: We appreciate all the reviewer's previous comments.

Reviewer #3 (Remarks to the Author):

In the present manuscript, the authors have made an effort to reveal how the emergence of weedy rice in tropical Asian regions may be strongly influenced by hybridization with wild species. While I believe that the topic of the manuscript is valuable and helps us understand the evolution of weediness in weedy-type crops, I think the manuscript still has some significant limitations.

Reply: We appreciate the reviewer's critique and comments. Please see our point-by-point responses to the three comments below.

Comment 1:

Firstly, I agree with the main conclusion that "wild rice introgression confers adaptability to Southeast Asian weedy rice," which has been reported many years ago. Unfortunately, this manuscript does not present any new findings or mechanisms for updating the evolution of Southeast Asian weedy rice.

Reply to the reviewer: As noted by Reviewer 1 above, the new genome sequence analyses in this study provide a valuable step forward. Evolutionary mechanisms underlying the origin and adaptation of Southeast Asian weedy rice have been proposed by some of our previous studies (*i.e.*, Song *et al.*, 2014, *Mol. Ecol.*; Qi *et al.*, 2015, *Mol. Ecol.*; Cui *et al.*, 2016, *G3*). However, the genome architecture and selective signatures of these hybrid-derived weedy strains remained unclear. Here our results indicate that these

Southeast Asian weedy rice strains show high genome dynamics. Candidate genes under selection are functionally related to abiotic stress, disease resistance and development. Thus, our new study provides important genomic insight into evolution of Southeast Asian weedy rice via wild rice introgression. Per the reviewer's comment, we have clarified the conceptual advance represented by our work in the revised version of the manuscript (Page 4, Lines 111-112).

Reviewer's feedback: Thank you for your reply. The genome architecture and selective signatures considering as the 'important genomic insight into evolution' mentioned by author are just my main concern. Because candidate genes derived from selection sweep and annotation, without any biological experimental evidence, can be uncertain and should not be used as a conclusion, especially with a small sample size in population genetics. I very much look forward to the authors providing new data or evidence to refute your comment. However, I noticed that the author only added a brief sentence in the introduction in response to my comment, stating that "the genome architecture and selective signatures of tropical Asian weedy strains remain largely unclear."

Reply to the reviewer's feedback: We appreciate this critical comment! We agree with the reviewer that a large number of weedy rice samples together with biological experimental data of the identified candidate genes will provide comprehensive evidence for the evolution of SEA weedy rice. However, the aims of our study are to: (1) reveal genome architecture and selection signatures of SEA weedy rice; and (2) address how adaptive introgression from wild rice overcomes the "cost of domestication" in weedy rice. As shown in results, our population genomic inferences based on worldwide rice accessions not only revealed how genetic introgressions from wild rice have shaped genome architectures of SEA weedy rice but also identified some candidate genes of selection in different weedy rice strains (Pages 4-5, Lines 115-134).

In particular, as shown in Supplementary Table 6, we identified causative alleles of some weediness traits in SEA weedy rice, such as seed shattering (*sh4* and *qSH1*), panicle architecture (*OsLG1*) and plant type (*prog1*), and awn length (*An-1* and *LABA1*). Molecular functions of these genes have been well documented in wild and cultivated rice. In our study, we discussed how crop and wild alleles of these domestication genes contributed to the formation of weediness traits in SEA weedy rice (Pages 8-10, Lines 241-289). For example, our previous study revealed that seed shattering has evolved independently in US-SH and US-BHA weedy rice through *de novo* mechanism (Li *et al.*, 2017, *Nature Genetics*). Here we showed that some SEA weedy rice acquired seed shattering trait from wild rice through genetic introgression. However, we also realized that we did not examine molecular functions of these newly identified candidate genes (shown in Supplementary Table 11). As Reviewer 1 commented, this is why we only briefly described in main text rather than emphasized this as a novelty of this study.

Per this comment, we re-wrote related content in the revised version of our manuscript (Page 4, Lines 111-114 and Page 5, Lines 123-134).

Comment 2:

Secondly, wild rice has good outcrossing habits, and gene exchange with neighboring rice is destined to happen. Although the authors succeeded in finding genomic evidence for a few cases by NGS, it cannot be regarded as a new rule or theory.

Reply to the reviewer: Weedy rice, like cultivated rice, is highly selfing. As such, introgression from outcrossing wild populations growing near rice fields is by no means a foregone conclusion. In the absence

of selection for adaptive introgression (as we have documented here), we suspect that hybridization rates would in fact be very low. More generally, understanding how genetic introgression has contributed to adaptation and speciation has long been a central issue in evolutionary biology. Here we identified adaptive introgressions from wild rice to weedy rice, as well as the genetic basis underlying the selection. Our newly framed Introduction (Page 2, Lines 32-33; Page 3, Lines 52-79; Page 11, Lines 318-327) now highlights this conceptual framework.

Reviewer's feedback: Thank you for your reply. It is common knowledge among rice-related researchers that even *temperate japonica* rice, which has the lowest stigma-exposed rate among ecotypes, can have an outcrossing rate of up to 5%. Weedy rice, *indica* rice, and wild rice have much higher stigma-exposed rates. While "destined to happen" may be an extreme phrase (due to certain reproductive isolation), it is reasonable to say that the emergence of a new ecotype of weedy rice is "not unexpected" due to this relatively high outcrossing rate. The authors' evidence for key domestication genes, such as *sh4*, supporting adaptive introgression from wild rice to weedy rice, is important. I understand and agree with your view, however, this evidence may not be exciting or innovative enough for potential readers in evolutionary biology and crop research. Regarding the newly framed Introduction, I agree that it effectively highlights the conceptual framework and increases the quality and value of the manuscript. However, I also agree that experimental and analytical evidence are better than explanations and citations to the research papers.

Reply to the reviewer's feedback: Again, we agree with the reviewer that further experimental evidence could provide better understanding of the weedy rice evolution. However, as we explained in the above comment, molecular functions of these domestication genes (Supplementary Table 6) have been confirmed in cultivated and wild rice in previous studies. In addition, we also clarified in the above comment that our study mainly focuses on genome architecture and evolutionary mechanism underlying the adaptation of SEA weedy rice, rather than the identification of novel genes of the weediness traits.

For these new candidate genes that identified by genome-wide scanning in this study, we think that further studies focusing on biological functions of these candidate genes can provide molecular evidence on the adaptive evolution of weedy rice. Per this comment, we provide more clarification in the revised version of our manuscript (Page 12, Lines 381-386).

Comment 3:

Lastly, I am concerned about whether such a small sample size (13 Malaysian and 18 Thai strains) is sufficient to reflect the objective diversity and population genomic information for Southeast Asian weedy rice. Performing population genetics analysis with this sample size would result in significant randomness. I believe that the results would change significantly if additional samples were added or some samples were replaced. Therefore, it is impossible to convincingly determine the genetic basis of weediness in weedy rice if the selection signal changes leading to changes in candidate genes accordingly.

Reply to the reviewer: In our previous studies, we examined the origin of Southeast Asian weedy rice using large sample sets (*i.e.*, 133 weed accessions in *Vigueira et al.*, 2019, *Evol. Appl.*). In the present study, the weedy rice accessions were selected to represent Southeast Asian weedy rice based on their genetic background and morphological traits. Per the reviewer's comment, we have clarified the criteria for choosing the accessions in the revised version of our manuscript (Pages 12-13, Lines 385-388).

Reviewer's feedback: Thank you for your reply. I agree that the weedy rice accessions selected based on

their genetic background and morphological traits can represent Southeast Asian weedy rice. However, the evasive answer still fails to address my concern effectively about whether the results would significantly change if additional samples were added or some samples were replaced. In population genetics, sample size determines accuracy to some extent based on the basic principles of statistics. While I agree with the representativeness of the sample, I have doubts about whether a small number of representative samples can reflect statistical accuracy enough to obtain general rules. Specifically, I am curious whether the selection sweeps for a small number of samples and a large number of samples overlap in this study. Whether or not there is an overlap is not an issue for me, but I regret that the author did not attempt to investigate this further.

Reply to the reviewer's feedback: We and the reviewer are in complete agreement on this most critical point: *"The weedy rice accessions selected based on their genetic background and morphological traits can represent Southeast Asian weedy rice."* We do not claim that this study provides a final, definitive picture of weedy rice evolutionary dynamics in Southeast Asia. But we do have statistically robust support for the specific conclusions that we draw relating to the role of wild rice adaptive introgression in the evolution of these tropical weed populations. In our previous studies, we showed that weedy rice was domesticated from cultivated rice at distinct domestication stages (*i.e.*, Song *et al.*, 2014, *Molecular Ecology*; Li *et al.*, 2017, *Nature Genetics*; Qiu *et al.*, 2020, *Genome Biology*). Here our genomic inferences based worldwide rice accessions further revealed that genetic introgressions from wild rice have contributed to the adaptive evolution of SEA weedy rice. We believe that the major conclusions of our study are reliable and that they represent a substantive advance in the understanding of evolutionary dynamics in crop-wild-weed species complexes.

Per the reviewer's comment, we provide more clarification in the revised version of our manuscript (Page 12, Lines 381-386).

Reviewer #4 (Remarks to the Author):

The revisions are satisfactory, congratulations.

Reply: We appreciate all the reviewer's previous comments.

Reviewers' Comments:

Reviewer #1:

Remarks to the Author:

I accept the corrections made, and I am satisfied with the manuscript.

Final comments on the outgroup genotyping and tree rooting:

From what I understand in your African rice genotyping, you only called positions that are variable in the 10 African accessions. The consequence of doing that is that in your merged (217 + 10) vcf file, you only have sites that are polymorphic in both datasets. That is not ideal, and possibly introduces some bias in the tree, PCA, and Treemix. For the purposes of outgroup-rooting, sites that are monomorphic in African rice and polymorphic in the 217 accessions would be actually more useful, because they allow to infer ancestral states (sites that are polymorphic in African rice are uninformative in this sense).

The procedure I had in mind is different - performing the mpileup for African rice, but specifying with the -l flag and a bed file the list of positions (from the 217-accession vcf file) that should be genotyped, whether or not they are polymorphic in the 10 African accessions (from the manual page: -l, --positions FILE: BED or position list file containing a list of regions or sites where pileup or BCF should be generated). I have done this in GATK (-L flag), but I am not sure how it works with mpileup.

Regarding the new Treemix figure that was shown in the rebuttal letter, I suspect there was an error in the command. The command should specify the outgroup (with the -root parameter), which should be the African rice. However, on the figure, African rice is an ingroup, i.e. it was not set as an outgroup. It is not clear which population was used as an outgroup.

Peter Civan, 21 Nov 2023

Reviewer #3:

Remarks to the Author:

I appreciate the efforts and replies from the authors. Their explanations and revisions have indeed improved the paper.

While the authors have emphasized that the study primarily focuses on the genome architecture and evolutionary mechanism underlying the adaptation of SEA weedy rice, rather than identifying novel genes for weediness traits, I agree with this perspective. It doesn't seem necessary to force the identification of novel genes for weediness traits.

However, I do have some concerns regarding the evidence supporting the claim of focusing mainly on genome architecture and evolutionary mechanism. The author did not provide a technical response to the technical question, the main concern that I raised (see comment 3), instead relying on opinions and quotations. While I still appreciate the main ideas and value of this paper, I am concerned about the limitations of the evidence supporting these ideas.

Reviewer #1 (Remarks to the Author):

I accept the corrections made, and I am satisfied with the manuscript.

Reply: We appreciate all the reviewer's comments throughout the review process, which have substantially improved our paper, and we are pleased that the reviewer is satisfied with the manuscript. We address the reviewer's additional final comments below:

Final comments on the outgroup genotyping and tree rooting:

From what I understand in your African rice genotyping, you only called positions that are variable in the 10 African accessions. The consequence of doing that is that in your merged (217 + 10) vcf file, you only have sites that are polymorphic in both datasets. That is not ideal, and possibly introduces some bias in the tree, PCA, and Treemix. For the purposes of outgroup-rooting, sites that are monomorphic in African rice and polymorphic in the 217 accessions would be actually more useful, because they allow to infer ancestral states (sites that are polymorphic in African rice are uninformative in this sense). The procedure I had in mind is different - performing the mpileup for African rice, but specifying with the -l flag and a bed file the list of positions (from the 217-accession vcf file) that should be genotyped, whether or not they are polymorphic in the 10 African accessions (from the manual page: -l, --positions FILE: BED or position list file containing a list of regions or sites where pileup or BCF should be generated). I have done this in GATK (-L flag), but I am not sure how it works with mpileup.

Reply: We appreciate this comment! As we clarified in previous responses, African rice (285 Mb genome size for *O. glaberrima* and 308 Mb for *O. barthii*) differs substantially in both the genome content and genome size to Asian rice (374 Mb for *O. sativa* ssp. *japonica*) (Stein *et al.*, 2018, *Nat. Genet.*), which complicates the unambiguous identification of orthologous fixed differences. In addition, the two VCF files (217 accessions of Asian rice and 10 accessions of African rice) were generated independently. Given that Asian rice is the reference genome for SNP calling, those loci that are not identified in African rice VCF file are likely either monomorphic or not present (due to the high genetic divergence between the two rice species) in African rice genome. This is why we only combined these loci that were identified as polymorphic in both VCF files. With very extensive re-analysis (essentially starting from scratch), we could redo all SNP calling to minimize the data bias. However, we do not think these new analyses would dramatically change the tree topologies, and moreover, tree rooting is not a major issue of our study. In particular, the wild Asian rice has long been used as outgroup for cultivated Asian rice (as we clarified in previous responses). We hope our responses and revisions (Page 13, Lines 389-395, 406-407, and 414-417; Page 14, 429-435) are able to address the reviewer's concern.

Regarding the new Treemix figure that was shown in the rebuttal letter, I suspect there was an error in the command. The command should specify the outgroup (with the -root parameter), which should be the African rice. However, on the figure, African rice is an ingroup, i.e. it was not set as an outgroup. It is not clear which population was used as an outgroup.

Reply: We appreciate this comment. Actually, we have attempted to perform *Treemix* with multiple different parameters. As shown below, even when we used the parameter "-root" to define African rice as outgroup, the tree topologies are still not reasonable. This is why we did not include the new *Treemix* results in the revised version of our manuscript.

Reviewer #3 (Remarks to the Author):

I appreciate the efforts and replies from the authors. Their explanations and revisions have indeed improved the paper.

Reply: We appreciate all the reviewer's comments, which have largely improved our manuscript.

While the authors have emphasized that the study primarily focuses on the genome architecture and evolutionary mechanism underlying the adaptation of SEA weedy rice, rather than identifying novel genes for weediness traits, I agree with this perspective. It doesn't seem necessary to force the identification of novel genes for weediness traits.

Reply: Yes, as the reviewer mentioned, our study aims to elucidate genome architecture and evolutionary mechanisms underlying the adaptation of SEA weedy rice. We and the reviewer are now in agreement on this point.

However, I do have some concerns regarding the evidence supporting the claim of focusing mainly on genome architecture and evolutionary mechanism. The author did not provide a technical response to the technical question, the main concern that I raised (see comment 3), instead relying on opinions and quotations. While I still appreciate the main ideas and value of this paper, I am concerned about the limitations of the evidence supporting these ideas.

Reply: Again, we agree with the reviewer that a large number of weedy rice samples will provide comprehensive evidence. However, as we clarified in our previous responses, we have investigated genetic background and morphological diversity of SEA weedy rice based on a large sample size. In this study, these SEA weedy rice accessions were selected based on their genetic background and morphological traits. Per this comment, we added more details in sections *Introduction* (Page 4, Line 115),

Discussion (Page 12, Lines 377-382) and *Materials & Methods* (Page 13, Lines 395-399). We hope our responses and revisions are able to address the reviewer's concern.